# Prognostic Implications of High-Degree Atrio-Ventricular Block in Patients with Acute Myocardial Infarction in the Contemporary Era

**DOI:** 10.3390/jcm12144834

**Published:** 2023-07-22

**Authors:** Jesús Velásquez-Rodríguez, Lourdes Vicent, Felipe Díez-Delhoyo, María Jesús Valero Masa, Vanesa Bruña, Iago Sousa-Casasnovas, Miriam Juárez-Fernández, Francisco Fernández-Avilés, Manuel Martínez-Sellés

**Affiliations:** 1Department of Cardiology, Hospital Universitario Severo Ochoa, 28914 Leganés, Spain; jesus1286@gmail.com; 2Department of Cardiology, Hospital Universitario 12 de Octubre, 28041 Madrid, Spain; mlourdesvicent@gmail.com (L.V.); felipediezdelhoyo@hotmail.com (F.D.-D.); vane_brufer@hotmail.com (V.B.); 3Department of Cardiology, Hospital General Universitario Gregorio Marañón, 28007 Madrid, Spain; mjvalero26@gmail.com (M.J.V.M.);; 4Facultad de Medicina, Universidad Complutense Madrid, 28040 Madrid, Spain

**Keywords:** myocardial infarction, prognosis, high-degree atrioventricular block

## Abstract

Background: High-degree atrioventricular block (HAVB) is a known complication of ST-segment elevation myocardial infarction (STEMI). We aimed to determine the prevalence and prognostic impact of HAVB in a contemporary cohort of STEMI. Methods: Data were collected from the DIAMANTE registry that included STEMI patients admitted to our cardiac intensive care unit treated with urgent reperfusion. We studied the clinical characteristics and evolution in patients with and without HAVB at admission. Results: From 1109 consecutive patients, HAVB was documented in 95 (8.6%). The right coronary artery was the culprit vessel in 84 patients with HAVB (88.4%). The independent predictors of HAVB were: male sex (OR 1.9, 95% CI 1.2–2.9), age (OR 1.03, 95% CI 1.01–1.05), involvement of right coronary artery (OR 12.4, 95% CI 7.6–20.2), and creatinine value (OR 1.5, 95% CI 1.1–2.0). A transient percutaneous pacemaker was used in 37 patients with HAVB (38.9%). Patients with HAVB had higher mortality that patients without HAVB (15.8% vs. 4.1%, *p* < 0.001); however, in multivariate analysis, HAVB was not an independent predictor of in-hospital mortality. Conclusions: HAVB was seen in 9% of STEMI patients and was particularly frequent in elderly males with renal failure. Patients with HAVB had a poor prognosis during hospitalization, but HAVB was not an independent predictor of in-hospital mortality.

## 1. Introduction

High-degree atrioventricular block (HAVB) is a known complication of ST-segment elevation myocardial infarction (STEMI). HAVB incidence has been reported in 1.5–13% of patients suffering STEMI [1,2,3,4,5,6,7,8,9,10]. Despite the improvements in reperfusion therapy, HAVB could be associated with an adverse prognosis. The majority of studies addressing the impact of HAVB were conducted in the pre-thrombolytic era and when primary percutaneous coronary intervention was uncommon [9,10,11,12,13]. Recent data regarding incidence, predictors and outcomes of HAVB in modern cohorts have shown discordant results regarding its prognostic impact. Some studies identified HAVB as an independent predictor of mortality [1,6,8] and others did not [4,7].

Our aim was to determine the prevalence and prognostic impact of HAVB at admission in patients with acute ST-elevation myocardial infarction (STEMI).

## 2. Materials and Methods

Retrospective analysis of the DIAMANTE (Descripción del Infarto Agudo de Miocardio: Actuaciones, Novedades, Terapias y Evolución—Description of Acute Myocardial Infarction: Management, New Therapies, and Evolution) registry. The methods of DIAMANTE were previously published [14,15,16,17]. This database included all patients with STEMI admitted to our cardiac intensive care unit.

Eligible patients were 18 years of age or older who had a STEMI diagnosis performed by a cardiologist, according to the presence of chest pain and ST-segment elevation [18,19]. All patients underwent urgent reperfusion therapy, including pharmacological fibrinolysis (the fibrin-specific agent tenecteplase was the most common agent), or primary percutaneous coronary intervention. In case of failed fibrinolysis (ST-segment resolution < 50% at 60–90 min; absence of typical reperfusion arrhythmias and/or chest pain relief), emergent rescue percutaneous coronary intervention was performed. According to the “pharmacoinvasive” strategy currently recommended [19], an early coronary angiography after successful fibrinolysis was routinely conducted within the first 24 h. Exclusion criteria were as follows: patients presenting later than 24 h of symptom onset or who did not undergo any reperfusion therapy, out-of-hospital cardiac arrest, participants who required endotracheal intubation prior to hospital arrival, and patients with non-obstructive coronary artery disease and no evidence of cardiac emboli as the cause of the STEMI (e.g., Takotsubo syndrome or coronary vasospasm). Patients with a history of prior AV conduction abnormalities, and cardiac device carriers who were dependent on pacemaker stimulation, were excluded from the study.

Definitions and endpoints: HAVB was defined as a third degree or second degree (Mobitz type 2) atrioventricular block. The primary endpoint was death from any cause during hospitalization. Deaths from any cause during follow-up were also recorded.

Continuous variables are presented as means (±standard deviation) and categorical variables are presented as frequencies and %. Comparisons between groups were made using Student’s *t*-test, or the nonparametric Mann–Whitney *U* test when appropriate, for continuous variables and the chi-square test for categorical variables. Additionally, odds ratio by logistic regression modelling was calculated for dichotomous variables to describe the strength of the relationship between the categorical risk factors and HAVB. Multivariate logistic regression models were developed to explore the relative contributions of the various risk factors. To determine the potential impact of HAVB on prognosis over time, we also analyzed survival effects with a Kaplan–Meier and Cox regression analysis. To determine which variables were entered into the final model, univariate comparisons were carried out comparing patients who were alive at the end of follow-up, and those who were not. The following variables were entered in the multivariate model when a *p* value < 0.10 was obtained in univariate analysis: age, body mass index, sex, hypertension, diabetes, hypercholesterolemia, smoking, peripheral arterial disease, chronic kidney disease, previous history of atrial fibrillation, previous stroke/transient ischemic attack, anticoagulation treatment, previous coronary artery bypass graft surgery, chronic pulmonary obstructive disease, anemia, active cancer, chronic heart failure, basic activities of daily living dependence, systolic/diastolic blood pressure and heart rate on arrival, infarct location, time-to-treatment, Killip class at presentation, angiography approach (radial or femoral), early VF (0–24 h), acute atrioventricular block, type of reperfusion therapy, preprocedural and postprocedural TIMI (thrombolysis in myocardial infarction) flow, multivessel disease, procedural success, type and use of stent, serum creatinine and hemoglobin levels, systolic dysfunction (considered to be present if left ventricular ejection fraction was <50%), presence of pericardial effusion, right ventricular dilatation/dysfunction, and mitral insufficiency (grades 0–4) on echocardiography (performed within the first 24 h of hospitalization).

Statistical analysis was performed with the SPSS 20.0 statistical package (IBM Corp., Armonk, NY, USA).

## 3. Results

A total of 1109 patients were included (Figure 1). The mean age was 64.1 ± 14.0 years, and 257 were females (23.2%). Primary percutaneous coronary intervention was performed in 1032 (93.1%) (including 39 with rescue percutaneous coronary intervention) and 77 (6.9%) received thrombolytic therapy alone. HAVB was documented in 95 patients (8.6%).

Baseline characteristics according to the incidence of HAVB are depicted in Table 1. HAVB patients were older, more frequently women, and presented comorbidities (atrial fibrillation, chronic heart failure, active cancer) and dependency more frequently than patients without HAVB. Patients with HAVB had a longer hospital stay and more post-infarction complications: cardiogenic shock, atrial fibrillation, ventricular arrhythmias, pericardial effusion, major bleeding, acute renal failure, infection, and higher mortality. Major bleeding was similar in both groups. There were no significant differences in the incidence of HAVB in patients treated with fibrinolysis compared to those treated with primary percutaneous coronary intervention. Table 2 presents coronary angiography findings. The right coronary artery was the culprit vessel in 88% of HAVB patients. Compared with patients without HAV, those with HAVB had a higher incidence of right ventricular infarction. The independent predictors of HAVB by multivariable analyses were: right coronary artery culprit lesion, male sex, creatinine value, and age (Table 3).

In patients with HAVB, a transient percutaneous pacemaker was used in 37 (39.0%). Compared to patients not treated with transient percutaneous pacemaker, these patients were older (74.8 ± 14.3 years vs. 68.9 ± 13.5, *p =* 0.04), had a longer time since symptom onset (6.1 ± 5.6 h vs. 3.6 ± 2.2 h, *p* < 0.001), and Killip class >II more frequently (64.5% vs. 36.2%, *p* = 0.006). Three of four patients with left anterior descending artery involvement (75.0%) required temporary pacemaker, in comparison with 40.4% of patients with right coronary artery as the culprit vessel (*p* < 0.001). When the left circumflex artery was the culprit vessel, the percutaneous pacemaker was not needed. Patients with a temporary pacemaker also developed heart failure during admission more frequently (21 [57.76%] vs. 16 [27.59%], *p* = 0.004) and had higher in-hospital mortality (11 [29.73%] vs. 4 [6.90%], *p* = 0.004). Patients who discharged with a permanent pacemaker (3, 8.1%) had right coronary artery as the culprit vessel.

A total of 58 patients (5.2%) died during hospitalization. The predictors of in-hospital mortality are presented in Table 4 and the predictors of long-term mortality in Table 5. HAVB was not independently associated with in-hospital mortality.

Nine patients were missing at the 30-day follow-up (0.8%), and long-term follow-up data could not be recorded in 22 patients (2.0%). The mean follow-up was 23.8 ± 19.4 months. A total of 147 patients (13.3%) died at the end of follow-up. HAVB had no effect on 30-day mortality. Long-term mortality was higher in patients with HAVB although this was mainly due to their high risk profile (hazard ratio 1.48, 95% confidence interval 0.51–4.36, *p =* 0.48) (Figure 2).

## 4. Discussion

In our contemporary cohort of STEMI treated with urgent reperfusion, HAVB occurred in 9% of the patients and was particularly frequent in elderly males with renal failure. The right coronary artery was involved in almost 90% of the cases. Thirty-nine percent of patients with HAVB required temporary pacing. Patients with HAVB had a poor in-hospital and long-term prognosis that was mainly due to the association of HAVB with age and comorbidity, as HAVB was not an independent predictor of mortality.

Early reperfusion, especially primary percutaneous coronary intervention, reduces the size of the infarct and decreases the impact and incidence of HAVB [1]. Different studies described the evolution of the incidence of atrioventricular block over time [9,10,20]. Spencer et al. [11] reported an incidence of complete atrioventricular block of 6% in the pre-thrombolytic era and 3% in the thrombolytic era. Similarly, Nguyen et al. [20] reported a decrease in the incidence from 5% in 1975 to 2% in 2005. Currently, in the middle of primary angioplasty era, the incidence of HAVB in several studies is less than 4% [1,3,4,5,6,7,8,9], except in the study Gómez-Talavera et al. [2], which found 12.6% of all degrees of AV block in STEMI patients, of which 8% had complete atrioventricular block. The rate of HAVB we found (8.7%) can be considered high and we consider that these results are explained by the prospective character of the DIAMANTE registry and, possibly, the very high percentage of patients with initial TIMI 0 flow compared with other studies [4,7].

Appendix A shows the comparison with previous series. We only included STEMI patients, while other series focused in all types of acute coronary syndromes [1,6] or only included patients with complete atrioventricular block [6,10]. HAVB can be seen in STEMI patients with inferior or anterior location, although it is more common in patients with inferior STEMI. The exact mechanism underlying atrioventricular block in STEMI remains unclear. However, there are several hypotheses depending on the location. In inferior STEMI, the first mechanism involves cardioinhibitory reflexes (Bezold–Jarisch) arising from vagal efferent arm in the ischemic left ventricular inferoposterior myocardial wall [21] and the second involves the effects of ischemia in the atrioventricular node [22], due to the involvement of the artery that irrigates the atrioventricular node distally from the posterolateral branch of the right coronary artery. In patients with left-dominance circulation, the atrioventricular node branch depends on the left circumflex artery. HAVB in anterior STEMI might result from extensive myocardial damage affecting the bundle branch traveling within the interventricular septum [7]. Therefore, HAVB in anterior STEMI is often preceded by bundle branch block, with an unstable escape rhythm [7].

In our study, there was no significant difference in the incidence of HAVB between patients treated with primary percutaneous coronary intervention and thrombolysis. HAVB was associated with poor revascularization results: 69.5% of patients with HAVB at admission had final TIMI III flow in comparison with 84.9% of patients without HAVB treated with percutaneous coronary intervention. This finding is consistent with the results of the Harmonizing Outcomes with Revascularization and Stents in Acute Myocardial Infarction (HORIZONS-AMI) trial [5], in which TIMI 0/I flow was an independent predictor of the development of HAVB, a marker of poor prognosis.

Regarding the need for pacemakers, the 2017 ESC Guidelines for the Treatment of Acute Myocardial Infarction in Patients with ST-segment Elevation [19] recommend temporary pacing in cases of sinus bradycardia with hemodynamic intolerance or HAVB without stable escape rhythm unresponsive to positive-chronotropic drugs. In our study, two fifths of our HAVB patients required a temporary pacemaker insertion, but only 3% needed permanent pacemaker implantation. These results are consistent with previous data [1,6,23]. Patients who required a temporary pacemaker had a longer delay in receiving reperfusion treatment; in addition to the infarct evolution time, the placement of a temporary pacemaker itself might increase the time delay in the catheterization laboratory. Patients with temporary pacemakers died more than those who did not require temporary pacing. The highest mortality in this group could be attributable to a more extensive myocardial ischemia [1]. We did not find significant differences in the incidence of possible complications related to pacemaker implantation (major bleeding, pericardial effusion, infections) regarding patients who did not require temporary pacing. HAVB was transient and reversible in all patients in whom the left anterior descending coronary artery was the culprit vessel. The resolution of HAVB after percutaneous coronary intervention in our patients suggests that the conduction block could be attributable to ischemia rather than necrosis of the intracardiac conduction system [24].

The presence of HAVB in patients with STEMI is considered an unfavorable prognostic marker. In-hospital mortality rates are significantly higher compared to patients without HAVB; in our study, for example, in-hospital mortality in STEMI with HAVB was 16%. Patients with HAVB had a poor in-hospital outcome, including a high risk of cardiogenic shock, left ventricular dysfunction, right ventricular infarction, and ventricular arrhythmias. Although HAVB used to be identified as an independent predictor of mortality [1,6,8], in our study, this was not the case, in line with recently published series [4,7,25]. Probably this is related to the fact that HAVB is not directly responsible for the increase in mortality, being just a marker of a large infarct size and a risky baseline profile. Our findings support this hypothesis, as patients with HAVB more frequently presented biventricular systolic dysfunction and cardiogenic shock.

In the HAVB group, the mortality was higher in anterior compared with inferior STEMI, as in previous publications [3,7]. This is most likely explained by more extensive infarctions when left anterior descending artery is the culprit lesion [3], and, possibly, the differences in the underlying pathophysiology of atrioventricular block in anterior and inferior STEMI as discussed above.

Regarding the impact HAVB on long-term prognosis both in the thrombolytic era and in the angioplasty era, the data are scarce and the results in some studies were discordant [7,11,26]. In DIAMANTE, patients with HAVB had a low long-term survival, but HAVB was not an independent predictor of long-term mortality, which is consistent with the results of Kim et al. [7].

Observational studies are vulnerable to selection bias and unidentified confounding factors, and so, our study had some limitations that must be recognized. As in any registry, the fact that some complications were not reported cannot be ruled out. Furthermore, we only included patients with HAVB at the time of admission, excluding those who could develop HAVB during hospitalization. We did not assess the time of HAVB occurrence or the details related to the implantation and time to explant of temporary pacemakers. Another limitation was the absence of peak values of cardiac biomarkers. We did not carry out a detailed analysis of the technical aspects of the coronary angiography and revascularization procedure that could impact the final vessel flow.

## 5. Conclusions

In a modern cohort of STEMI treated with urgent reperfusion, HAVB occurred in 9% of the patients and was particularly frequent in elderly males with chronic kidney disease. The right coronary artery was involved in almost 90% of HAVB cases. HAVB was not an independent predictor of in-hospital mortality.

## Figures and Tables

**Figure 1 jcm-12-04834-f001:**
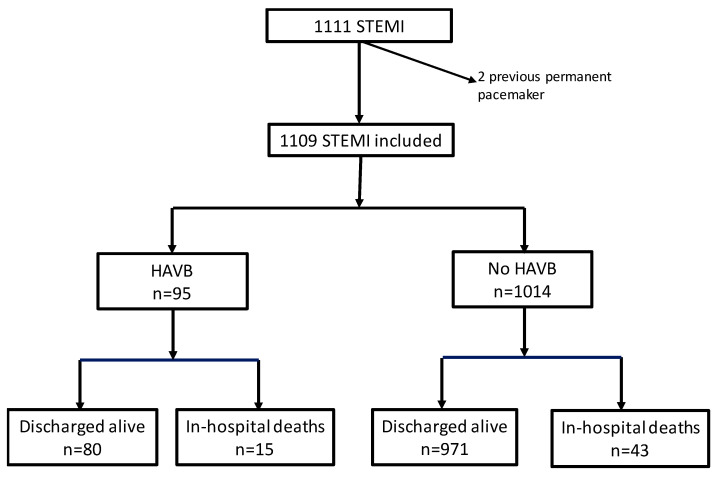
Flow chart of the enrollment of the study participants. STEMI: ST-segment elevation myocardial infarction. HAVB High degree atrioventricular block.

**Figure 2 jcm-12-04834-f002:**
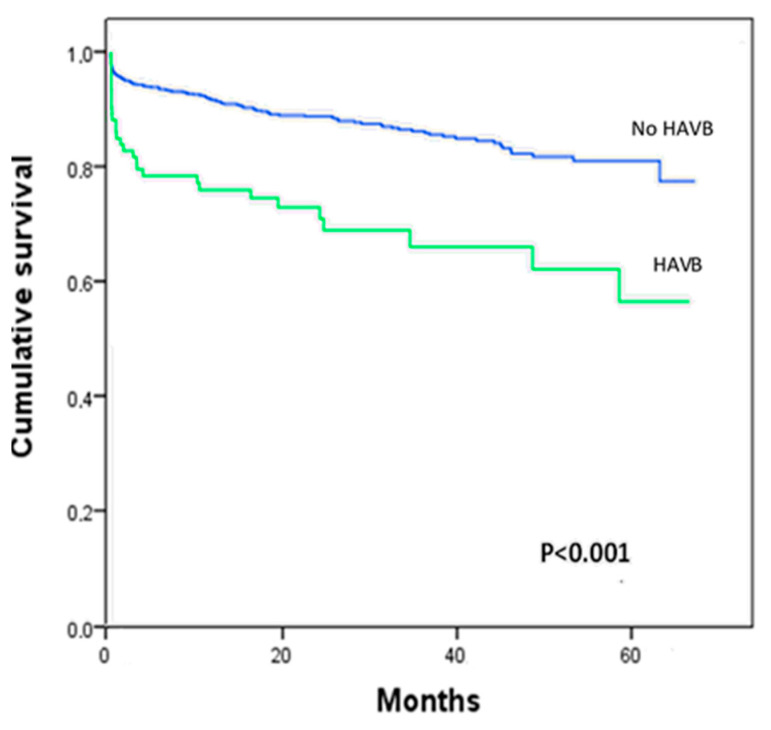
Cumulative survival according to the presence of high-degree atrioventricular block (HAVB).

**Table 1 jcm-12-04834-t001:** Baseline characteristics.

	No HAVB (n = 1014)	HAVB (n = 95)	*p*
Age, years ± SD	63.4 ± 13.8	70.9 ± 14.3	0.23
Female sex	224 (20.2%)	33 (34.7%)	0.005
Hypertension	543 (53.6%)	58 (61.1%)	0.16
Diabetes mellitus	210 (20.1%)	27 (28.4%)	0.08
Dyslipidemia	466 (45.9%)	46 (48.4%)	0.61
Active smoker	458 (45.2%)	34 (35.8%)	0.08
Body mass index	27.8 ± 4.4	27.1 ± 4.0	0.93
Chronic Obstructive Pulmonary disease	71 (7.0%)	8 (8.4%)	0.61
Previous atrial fibrillation	35 (3.5%)	12 (12.6%)	<0.001
Chronic heart failure	35 (3.5%)	12 (12.6%)	<0.001
Previous cardiac surgery	86 (8.5%)	9 (9.5%)	0.001
Chronic kidney disease	67 (6.6%)	14 (14.7%)	0.004
Peripheral artery disease	46 (4.5%)	2 (2.1%)	0.27
Dependent	25 (2.5%)	8 (8.4%)	0.001
Active cancer	86 (8.5%)	9 (9.5%)	0.006
Infarct site			
Anterior	452 (44.6%)	5 (5.3%)	<0.001
Inferior, lateral or posterior	560 (55.2%)	89 (93.7%)	
Left bundle branch block	2 (0.2%)	1 (1.0%)	
Hours to reperfusion	4.5 ± 3.9	4.6 ± 4.1	0.96
Creatinine (g/dL)	0.97 ± 0.52	1.30 ± 0.84	<0.001
Systolic blood pressure (mmHg)	135.3 ± 29.1	105.8 ± 27.1	0.79
Diastolic blood pressure (mmHg)	77.6 ± 17.6	61.5 ± 15.1	0.10
Left ventricular ejection fraction (%)	46.0 ± 12.1	46.3 ± 11.3	0.26
Ventricular fibrillation	72 (7.1%)	18 (18.9%)	<0.001
Killip > II	185 (18.2%)	45 (47.4%)	<0.001
Hospital stay (days)	6.8 ± 14.1	10.3 ± 18.9	0.006
Complications			
Cardiogenic shock	137 (13.5%)	37 (38.9%)	<0.001
Right ventricle infarction	60 (5.9%)	41 (43.2%)	<0.001
Atrial fibrillation post-STEMI	63 (6.2%)	13 (13.7%)	0.05
Ventricular arrhythmias post-STEMI	35 (3.5%)	10 (10.5%)	0.001
Pericardial effusion	23 (2.3%)	2 (2.1%)	0.93
Major bleeding	38 (3.7%)	7 (7.4%)	0.09
Acute kidney injury	88 (8.7%)	22 (23.2%)	<0.001
Infections	41 (4.0%)	13 (13.7%	<0.001
In-hospital death	43 (4.2%)	15 (15.8%)	<0.001
Treatment			
Fibrinolysis	107 (10.6%)	13 (13.7%)	0.26
Radial access	757 (74.7%)	41 (43.2%)	<0.001
Complete revascularization at discharge	771 (76.0%)	65 (68.4%)	0.09
Temporary pacemaker	--	37 (38.9%)	--
Permanent pacemaker	--	3 (3.2%)	--
Betablocker at discharge	842 (83.0%)	51 (53.7%)	<0.001
ACE inhibitors at discharge	832 (82.1%)	62 (65.3%)	0.04

SD = standard deviation; STEMI = ST-segment elevation myocardial infarction; HAVB: high-degree atrioventricular block. ACE: Angiotensin convertor enzyme.

**Table 2 jcm-12-04834-t002:** Coronary angiography findings and invasive management.

	No HAVB (n = 1014)	HAVB (n = 95)	*p*
Left main or three vessels disease	175 (17.3%)	17 (17.9%)	0.89
Location of culprit lesion			
Left main	6 (0.6%)	0	<0.001
Descending anterior artery	461 (45.5%)	4 (4.2%)	
Circumflex artery	165 (16.3%)	7 (7.7%)	
Right coronary artery	382 (37.8%)	84 (88.4%)	
Initial TIMI flow			
0	705 (69.5%)	74 (77.9%)	0.29
I	80 (7.9%)	7 (7.7%)	
II	75 (7.4%)	3 (3.2%)	
III	154 (15.2%)	11 (11.6%)	
Type of stent			
Bare metal stent	339 (33.4%)	45 (47.4%)	0.01
Drug eluting stent	672 (66.2%)	50 (52.6%)	
Final TIMI 3 flow	861 (84.9%)	66 (69.5%)	<0.001

TIMI = thrombolysis in myocardial infarction.

**Table 3 jcm-12-04834-t003:** Independent predictors of high-degree atrioventricular block.

	Odds Ratio	Confidence Interval 95%	*p*
Right coronary artery culprit lesion	12.41	7.61–20.21	<0.001
Male sex	1.87	1.20–2.93	<0.001
Creatinine (mg/dL)	1.45	1.07–1.97	0.001
Age	1.03	1.01–1.05	0.001

**Table 4 jcm-12-04834-t004:** Independent predictors of in-hospital mortality.

	Adjusted OR	*p* Value Adjusted OR	Crude OR	In-Hospital Mortality
Killip II-IV	5.4 (2.2–13.6)	<0.001	27.6 (13.3–57.1)	22.2%
Hemoglobin (g/dL)	0.8 (0.6–0.9)	0.01	0.65 (0.57–0.74)	19.3% ^1^
Age (years)	1.04 (1.00–1.07)	0.03	1.07 (1.05–1.10)	12.2% ^2^
Left ventricular ejection fraction (%)	0.93 (0.90–0.96)	<0.001	0.89 (0.87–0.92)	14.8% ^3^
Significant pericardial effusion	7.9 (2.0–31.7)	0.004	10.5 (4.3–25.7)	32.0%
Ventricular arrhythmias	7.4 (2.9–18.4)	<0.001	26.5 (13.6–51.8)	48.9%
Final TIMI 3 flow	0.28 (0.13–0.61)	0.001	0.14 (0.08–0.25)	2.9%
Chronic kidney disease	3.8 (1.5–9.9)	0.005	6.1 (3.3–11.3)	21%
HAVB	1.48 (0.51–4.36)	0.47	---	15.8%

TIMI = thrombolysis in myocardial infarction; HAVB = high-degree atrioventricular block; OR = odds ratio. ^1^ = Hemoglobin < 12 g/dL. ^2^ = Age > 75 years. ^3^ = Left ventricular ejection fraction < 40%.

**Table 5 jcm-12-04834-t005:** Independent predictors of long-term mortality.

	Adjusted HR	*p* Value Adjusted HR	Crude HR	Long-Term Mortality
Killip II-IV	2.0 (1.3–2.9)	0.001	5.8 (4.2–8.0)	35.7%
Hemoglobin (g/dL)	0.85 (0.78–0.92)	<0.001	0.65 (0.57–0.74)	36.7% ^1^
Age (years)	1.05 (1.03–1.07)	<0.001	1.07 (1.05–1.10)	28.7% ^2^
Left ventricular ejection fraction (%)	0.96 (0.95–0.98)	<0.001	0.89 (0.87–0.92)	25.3% ^3^
Ventricular arrhythmia	3.1 (1.9–5.0)	<0.001	26.5 (13.6–51.9)	62.2%
Final TIMI 3 flow	0.64 (0.44–0.94)	0.02	0.14 (0.08–0.25)	10.3%
Chronic kidney disease	2.5 (1.6–3.9)	<0.001	6.1 (3.3–11.3)	38.3%
HAVB	1.34 (0.69–2.60)	0.39	---	27.6%

HAVB: high-degree atrioventricular block; HR = hazard ratio; ^1^ = Hemoglobin < 12 g/dL; ^2^ = Age > 75 years; ^3^ = Left ventricular ejection fraction < 40%.

## Data Availability

The data that support the findings of this study are available on request from the corresponding author.

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
