# Peer review of "Prognostic Implications of High-Degree Atrio-Ventricular Block in Patients with Acute Myocardial Infarction in the Contemporary Era"

_jcm, 2023, doi:10.3390/jcm12144834_

Round 1
Reviewer 1 Report
The authors evaluated the impact of HAVB on the prognosis of STEMI patients based on a prospective cohort. I have some comments.
The major comments:
1. Abstract: The aim of this study was to determine the impact of HAVB on the prognosis of STEMI, so the results should report the effect of HAVB. I don't think the results fit the purpose.
2. Study design: Is this study designed at the beginning of the cohort or a retrospective analysis? Please make it clear. If this study is designed at the beginning, is it registered on any website?
3. Introduction: Too many aims are presented, It's highly recommended to define the primary objective.
4. Methods: The variables and the rule of selecting variables into the logistic and Cox regression models should be clearly stated. It's recommended to adjust as many confounding factors as possible. The significance level this study used and the reporting guidelines this study followed should be displayed.
5. Results: I recommend adding a flow chart of the enrollment of the study participants.
6. Results: For the analysis of Table 3, why are only 4 variables included in the model? From my point of view, it's an association analysis, and better removed from the results.
7. Tables 4 and 5: Explain why not reporting the crude OR and HR.
The minor comments:
1. Check whether reference 18 is appropriately referred.
2. The quality of Figure 1 should be improved to fit the publication.
Author Response
Response to Reviewer 1
We would like to thank the comments from reviewer 1 that have helped us to improve our manuscript.
Point 1: 1. Abstract: The aim of this study was to determine the impact of HAVB on the prognosis of STEMI, so the results should report the effect of HAVB. I don't think the results fit the purpose.
Response 1: We have changed the Abstract according to this comment.
Point 2: Study design: Is this study designed at the beginning of the cohort or a retrospective analysis? Please make it clear. If this study is designed at the beginning, is it registered on any website?
Response 2: This is a retrospective analysis of the DIAMANTE (Descripción del Infarto Agudo de Miocardio: Actuaciones, Novedades, Terapias y Evolución — Description of Acute Myocardial Infarction: Management, New Therapies and Evolution) registry. We have clarified this in Methods.
Point 3. Introduction: Too many aims are presented, It's highly recommended to define the primary objective.
Response 3: We have edited the last paragraph of the introduction to define the main objective of our study.
Point 4. Methods: The variables and the rule of selecting variables into the logistic and Cox regression models should be clearly stated. It's recommended to adjust as many confounding factors as possible. The significance level this study used and the reporting guidelines this study followed should be displayed.
Response 4: This has been clarified.
Point 5. Results: I recommend adding a flow chart of the enrollment of the study participants.
Response 5: We have included the flow chart in a new figure.
Point 6. Results: For the analysis of Table 3, why are only 4 variables included in the model? From my point of view, it's an association analysis, and better removed from the results.
Response 6: Table 3 shows the independent predictors of HAVB. These were the only variables that remained in the model after multivariable analysis.
Point 7: Tables 4 and 5: Explain why not reporting the crude OR and HR.
Response 7: OR for inhospital mortality and HR for long-term mortality. We have made the correction in the new version of the article.
The minor comments:
- Check whether reference 19 is appropriately referred.
Response: We have modified reference 19.
- The quality of Figure 1 should be improved to fit the publication.
Response: We have improved the figure quality.
Reviewer 2 Report
This retrospective paper on high-degree atrioventricular block (HAVB) in STEMI patients is well-written. I would like to extend my congratulations to the authors for their work. The importance of ECG as a diagnostic and prognostic tool in patients with acute coronary syndromes remains significant today as highlighted in this paper. I have a few minor comments to share.
1) The authors should provide an explanation regarding the proportion of patients (up to 20%) who underwent fibrinolysis. If this information is not included, it should be mentioned as a limitation of the study.
2) The authors should discuss the low rate of permanent pacemaker implantation in this population.
Quality of english adeguate
Author Response
Response to Reviewer 2.
We would like to thank the comments from reviewer 1 that have helped us to improve our manuscript.
This retrospective paper on high-degree atrioventricular block (HAVB) in STEMI patients is well-written. I would like to extend my congratulations to the authors for their work. The importance of ECG as a diagnostic and prognostic tool in patients with acute coronary syndromes remains significant today as highlighted in this paper. I have a few minor comments to share.
Response: Thank you for these comments.
1) The authors should provide an explanation regarding the proportion of patients (up to 20%) who underwent fibrinolysis. If this information is not included, it should be mentioned as a limitation of the study.
Response 1: We are sorry for this mistake. In the non-HAVB group 10.6% underwent fibrinolysis (not 19.7%). The proportion of patients undergoing fibrinolysis is explained by the fact that the first patients included date back to 2010.
2) The authors should discuss the low rate of permanent pacemaker implantation in this population.
Response 2: We have extendend the explanation on this aspect in the discussion.
Round 2
Reviewer 1 Report
The authors have addressed the majority of my questions, but in the previously published papers, there were 1111 patients included, while in this study it's 1109 patients. Transparency is my major concern. I wonder why the published papers no one clearly shows how many patients were screened and how many patients were excluded according to the exclusion criteria. That's why I recommended the authors add a flow chart.
Other remaining comments:
1. Explain why not reporting the crude OR and HR of HAVB in Table 4.
2. The background color of Figure 2 is better to remove.
Author Response
Response to Reviewer Comments
The authors have addressed the majority of my questions, but in the previously published papers, there were 1111 patients included, while in this study it's 1109 patients. Transparency is my major concern. I wonder why the published papers no one clearly shows how many patients were screened and how many patients were excluded according to the exclusion criteria. That's why I recommended the authors add a flow chart.
Response: In this study, the sample was 1109 patients, not 1111, because 2 patients with previously implanted permanent pacemakers were excluded. We have modified the algorithm of Figure 1 to clarify this aspect.
Other remaining comments:
- Explain why not reporting the crude OR and HR of HAVB in Table 4.
Response 1: We have reflected in the table only the adjusted value of OR or HR (not the crude OR o HR ratio) because it is with the adjusted ratio that the rest of the variables are taken into account in the multivariate analysis.
- The background color of Figure 2 is better to remove.
Reponse to point 2: We have removed the background color from Figure 2 in the new version of the manuscript.